# Application of Pseudoinfectious Viruses in Transient Gene Expression in Mammalian Cells: Combining Efficient Expression with Regulatory Compliance

**DOI:** 10.3390/biom15020274

**Published:** 2025-02-13

**Authors:** Gulzat Zauatbayeva, Tolganay Kulatay, Bakytkali Ingirbay, Zhanar Shakhmanova, Viktoriya Keyer, Mikhail Zaripov, Maral Zhumabekova, Alexandr V. Shustov

**Affiliations:** 1National Center for Biotechnology, 010000 Astana, Kazakhstan; zauatbaeva@biocenter.kz (G.Z.); kulatay@biocenter.kz (T.K.); ingirbay@biocenter.kz (B.I.); zhanar.shakhmanova@bk.ru (Z.S.); keer@biocenter.kz (V.K.); zhumabekova@biocenter.kz (M.Z.); 2Institute of Theoretical and Experimental Biophysics, 142290 Pushchino, Russia; mizaripov@mail.ru

**Keywords:** transient gene expression, mammalian expression, viral vector, alphavirus, Venezuelan equine encephalitis virus, replicon, pseudoinfectious virus, immobilized metal affinity chromatography, IMAC

## Abstract

Transient gene expression (TGE) is commonly employed for protein production, but its reliance on plasmid transfection makes it challenging to scale up. In this paper, an alternative TGE method is presented, utilizing pseudoinfectious alphavirus as an expression vector. Pseudoinfectious viruses (PIV) and a replicable helper construct were derived from the genome of the Venezuelan equine encephalitis virus. The PIV carries a mutant capsid protein that prevents packaging into infectious particles, while the replicable helper encodes a wild-type capsid protein but lacks other viral structural proteins. Although PIV and the helper cannot independently spread infection, their combination results in increased titers in cell cultures, enabling easier scale-up of producing cultures. The PIV-driven production of a model protein outperforms that of alphavirus replicon vectors or simple plasmid vectors. Another described feature of the expression system is the modification to immobilized metal affinity chromatography (IMAC), allowing purification of His-tagged recombinant proteins from a conditioned medium in the presence of substances that can strip metal from the IMAC columns. The PIV-based expression system allows for the production of milligram quantities of recombinant proteins in static cultures, without the need for complex equipment such as bioreactors, and complies with regulatory requirements due to its distinction from common recombinant viruses.

## 1. Introduction

A spectrum of mammalian expression systems has been developed to meet the needs of biomedical research and industry. Two approaches are in common use. Industry tends to use selected clones of cell lines stably transformed with a gene of interest (GOI) [1]. In academia, transient gene expression (TGE) is a commonplace, generally performed in cell cultures transfected with plasmid vectors [2]. However, no single approach may fit different demands considering a diversity of particular conditions and protein expression projects.

The path of creating stably transformed cell lines, then achieving amplification of the GOI copy number (e.g., using methotrexate or methionine sulfoximine selection) and selecting clones with high productivity is, in general, a long (6 months or more) labor-extensive endeavor, and may request impermissible resources from a research laboratory in academia [3]. There are situations when stable producers are impractical; for example, when a panel of purified products is tested for activity, whereas small amounts of each variant are required for testing, and the majority of expressed variants will not be developed further after completion of the testing. In such cases, transient gene expression (TGE) is preferred. TGE is frequently performed using plasmid vectors in which the GOI is placed under the control of a strong mammalian promoter [4].

However, TGE itself faces known problems, one being a rapid decline in the productivity of transfected cells because plasmid vectors present in the cells are diluted during successive cell divisions, and another being a phenomenon of “expression inactivation”, caused by methylation of promoters in extrachromosomal DNA by cellular methylases [5]. Among the main obstacles to the use of TGE for industrial production, is the need to perform large-scale DNA transfections. The process of transfection itself yields results that are too variable for industrial use. All methods of transfection appear sensitive to media, cell types, transfection reagents, and variables, such as the health of the culture, surface area/volume for adherent cells, and mixing time/agitation rate for suspension cultures [6]. Generally, transfections require new optimization each time the process set-up is changed, e.g., when ‘scaling up’. Entering the poorly mapped field of large-scale transfections from the settings of a research laboratory is risky, as there is no guarantee that the right parameters will be found, and the resulting process will suit demands.

Adding more hurdles to large-scale TGE are the need to obtain large quantities of transfection-grade plasmids, and the use of proportional amounts of transfection reagents. Economic considerations are also important, because commercial transfection reagents, such as cationic liposomes, being efficient in delivering plasmids, become prohibitively expensive when consumed in large quantities.

Other types of expression vectors, alternative to plasmids, are being developed in the field. For example, retroviral and lentiviral vectors capable of genomic integration are convenient tools for achieving stable genomic integration of the gene of interest (GOI). In contrast, integration-deficient lentiviral vectors (IDLVs), which carry a defect in the integrase gene, are suitable for generating so-called pseudostably transfected cell pools, which can serve as production cultures for generating moderate amounts of recombinant proteins. A review of reverse-transcription or DNA-virus vectors is outside the scope of this introduction.

Having had experience with large-scale transfections (e.g., transfection of 10–20 flasks T-175 in a batch) for research projects involving mammalian expression, authors of this study have formed the opinion that in the research laboratory, the results of large-scale transfections are too variable. To gain more control over the TGE’s results, we transitioned to a different technology for setting up TGE, including the use of high-producing RNA-vectors and infection with single-round infectious particles to generate producing cultures.

There are interesting alternatives for plasmid transfection-driven TGE. RNA molecules may be employed as expression vectors, and RNA molecules capable of autonomous replication in the cytoplasm are of particular interest for this study [7]. Such RNA vectors can be derived from the genomes of RNA-containing viruses [8]. The resulting RNAs replicate autonomously because they encode viral proteins constituting the replicase (including RNA-dependent RNA polymerase) and carry cis-acting sequences, which interact with the replicase to control RNA replication [9,10].

Cytoplasmic replication of expression vectors eases the dependence of an expression system on cell nuclear processes, and some RNA viruses block nuclear functions and usurp the translational machinery to translate only proteins encoded in the viral RNA. In conditions of TGE, this redirecting of cellular resources to only vector-driven translation can be beneficial.

Another feature of autonomously replicating RNA vectors is that, because such vectors are derived from viral genomes, they can be created to retain the capacity of packaging into infectious particles, which are morphologically similar to virions [11,12,13]. This enables us to package the expression vector into virions and use the resulting virus for infection to generate producing cultures. The advantage of the latter process is that infection is many orders of magnitude more efficient per delivered vector molecule than transfection, and infection is easily scalable.

Alphaviruses, members in the genus Alphavirus from the Togaviridae family, have been used to create autonomously replicating RNAs that have a variety of molecular designs [14,15,16,17]. Alphaviruses are widespread throughout the world, and the genus includes many species [18]. However, three particular species are more often used for genetic engineering; namely the Sindbis virus, Semliki Forest virus, and Venezuelan equine encephalitis virus (VEEV) [19,20].

The alphavirus genome is an 11.5 kb, single-stranded RNA of positive-sense (i.e., RNA is translated directly on the ribosomes) [9]. The RNA carries a 5′-methylguanylate cap and a 3′-polyadenylate tail. Untranslated regions 5′UTR and 3′UTR also serve as cis-acting sequences that interact with the viral replication machinery. Genomic RNA contains two long open reading frames (ORFs). The first ORF encodes nonstructural proteins nsP1–nsP4, which associate to form the viral replicase. The replicase provides for the genome replication and synthesis of a smaller-type viral RNA, subgenomic RNA, also of (+) polarity. The replicase transcribes the subgenomic RNA beginning from a region (dubbed the subgenomic promoter) within a minus-strand RNA intermediate. Subgenomic RNA (26S RNA) encompasses the second ORF, and the translation of subgenomic RNA gives rise to viral structural proteins (i.e., composing the virion) C-E3-E2-6k-E1. Among structural proteins, the capsid protein (C protein) encapsidates genomic RNA to form viral nucleocapsid; proteins E3-E2-6k-E1 are surface glycoproteins forming spikes on the surface of virions. Structural proteins are not required for the viral RNA replication in the cytoplasm.

One feature natural to alphaviruses and an attractive property for expression systems is the high replicative activity of alphaviruses, manifested in large titers of daughter virions (10^9^–10^10^ infectious particles per milliliter are typical), and high levels of intracellular viral RNAs (10^6^ copies per cell) and proteins [21]. Expression from GOI inserted into alphaviral RNA can result in 25% of the total cellular protein being the GOI product [22].

Alphaviral replicon is a fragment of the genome, which has cis-acting sequences 5′UTR and 3′UTR, encodes replicase (nsP1-nsP4), and contains no genes for structural proteins [23]. Most studies with alphavirus vectors have been conducted using replicons. Typically, a GOI is inserted in place of structural protein genes, resulting in the subgenomic RNA encoding the GOI product. Commonly used alphaviral replicons retain a packaging signal (which is located in nsP1 or nsP2 in different species). Replicons will be packaged into infectious particles (replicon particles) if all structural proteins C-E3-E2-6k-E1 are provided in trans (i.e., from different engineered constructs) [24]. A packaging helper is an RNA molecule designed to produce structural proteins, which are consumed for the encapsidation of the replicon RNA and the formation of virions [12,24].

The term “pseudoinfectious virus” is relatively new in the field. Specifically, modified alphavirus genomes described as pseudoinfectious viruses (PIVs) by their authors are introduced in [12]. Unlike replicons, which lack structural genes entirely, PIVs retain genes encoding all viral structural proteins. To create PIVs, modifications are introduced into the structural protein genes, preventing the packaging of the viral genome into infectious particles while still allowing the production of non-infectious (empty) virions, referred to as virus-like particles (VLPs). In principle, certain PIV variants may produce infectious particles during intracellular replication. However, in this case, the PIV generates low titers of infectious particles, insufficient to cause infection with increasing titers or to propagate the PIV in naïve cell cultures like a regular virus [12]. However, as in the case with replicons, PIVs produce infectious particles to high titers when the packaging defect is counteracted by providing functional packaging proteins in trans [12].

This work’s aim was to develop a technology for producing recombinant proteins in mammalian cell cultures given the following preconditions. The technology has to be suitable for research laboratories with relatively simple equipment, which does not include bioreactors. Also, it was dictated by the equipment availability that only adhesion-dependent cells growing in still cultures (flasks, cell factories) are employed in the production process. With this in mind, and remembering that DNA-transfection of many flasks previously produced unstable results for the authors, we elected to use infection rather than transfection, to generate large producing cultures.

For this study, two types of vectors were constructed based on the VEEV genome. The first type consists of well-known alphaviral replicons, which were designed to express GFP and SEAP as markers. The second type is a novel vector, the pseudoinfectious virus (PIV), which carries the same expression cassette with GFP and SEAP genes. Infectious particles were generated through helper-facilitated in-trans packaging using small-scale transfections, and these particles were used to infect large-scale producing cultures. The results indicate that PIV outperforms replicons in terms of production. The tagged secretory product SEAP was accumulated in amounts sufficient for convenient purification via metal affinity chromatography (IMAC), with final preparations containing >5 mg/mL of SEAP. Modifications to the IMAC protocol are described, allowing successful purification even when the conditioned medium contains substances that interfere with IMAC by stripping metal ions from the IMAC columns.

## 2. Materials and Methods

### 2.1. Cell Cultures

Cells BHK-21 (ATCC CCL-10) and HEK293FT (a fast-growing subclone of human embryonic kidney cells HEK293T, ThermoFisher, Waltham, MA, USA, Cat, R70007) were grown in a complete growth medium, which is Dulbecco’s Modified Eagle’s Medium (DMEM) with high glucose (Lonza Cat, BE12–604, Morristown, NJ, USA) with the addition of 10% fetal bovine serum (FBS, Gibco Cat, 16000–044, Amsterdam, The Netherlands), 2 mM L-glutamine (Lonza Cat, BE17–605E, Verviers, Belgium), 1% MEM vitamin solution (ThermoScientific Cat, 11120052, Gloucester, UK), 1% non-essential amino acids (ThermoScientific Cat, 11140050, UK), 100 U/mL penicillin, and 100 µg/mL streptomycin (Gibco Cat, 15140–122, Amsterdam, The Netherlands).

### 2.2. Gene Cassette for Expression of GFP and SEAP

A gene-encoding green fluorescent protein (GFP) was amplified from Addgene Cat. 113013. A gene for human-secreted alkaline phosphatase (SEAP) is from pVITRO1-neo-GFP/SEAP (InvivoGen, San Diego, CA, USA, Cat. pvitro1-ngfpsp). To create an expression cassette, ORFs for GFP and SEAP were joined in a contiguous ORF through the 2A-peptide (foot-and-mouth disease virus autoprotease 2A) as a linker. Kozak sequence (GCCGCCACC-ATGG, where ATG is the start codon) was engineered at the beginning of the gene cassette and this Kozak sequence is present in all expression constructs in this study. The SEAP has a signal peptide directing the protein into the secretory pathway. A histidine tag at the C-terminus of SEAP is ten residues-long, 10xHis. Mapped sequences of the cassette and expressed product are given in Appendix A.

### 2.3. DNA-Launched Replicons, Defective Helpers, and a Simple Expression Vector

The parental viral genome is VEEV strain TC-83 (Genbank Acc, EEVNSPEPB). Nucleotide positions reported herein are numbered as in the TC-83 genome. Numbering of a.a. residues is local and pertains to mature protein.

All viral constructs in this study were created in the form of molecular infectious clones (MICs). A MIC is a plasmid containing a full-length genome (autonomously replicating RNA) cloned with additional elements, which allow rescuing the RNA-genome replication after DNA-transfection. An overview of the molecular design of MIC is shown in Figure 1a.

The viral sequence is cloned under the control of the CMV promoter. Antigenomic ribozyme of hepatitis D virus and a strong transcriptional termination/polyadenylation signal (from the human growth hormone gene, HGH) are cloned downstream of the oligo-A stretch, which terminates the VEEV genome. After the initial transcription from transfected plasmid and export of the transcript to the cytoplasm, the RNA transcript begins autonomous replication mediated by the replicase proteins nsP1-nsP4 encoded in the RNA. This replication is no longer dependent on the nuclear transcription. Standard methods of genetic engineering were used. Sequences are available upon request.

To create replicons, the gene-encoding structural proteins were removed from the TC-83 genome and the gene cassette GFP-2A-SEAP was cloned in their place under the control of the subgenomic promoter. Two replicons were created, VEErepGS and VEErepGS.nsP2mut (Figure 1b,c), which differ in only one nucleotide position 3865 (wild type adenine in VEErepGS; or mutant thymidine in VEErepGS.nsP2mut). This change leads to Gln to Leu substitution in nsP2 at a.a. position 739. The mutation Q739L enables VEEV replicons to replicate persistently in cells that produce type I interferons (IFN-I, including alpha- and beta-IFN) in response to viral infection and establish unresponsiveness to viral infection (antiviral state) [25]. One cell line used in this work (HEK293FT) is IFN-competent (produces IFN-I and responds to IFN-I). The other line BHK-21 cannot produce IFN-I. The mutant replicon was created to test if the capacity to establish persistent infection (in IFN-competent cells) is advantageous to obtain higher expression levels.

Defective helpers (DHs) are fragments of the VEEV genome, which lack the functional replicase and hence cannot replicate on their own. However, DHs retain cis-acting sequences 5′UTR and 3′UTR and will replicate in cells possessing the functional VEEV replicase. DHs were created to retain one or more structural genes to be used as packaging helpers. Helpers DH-delC and DH-delE (Figure 1d–f) carry a deletion in nsP1-nsP4, which was originally tested in the work [26]. Because of the deletion, DHs encode only 158 a.a. from the N-terminus of nsP1 and 77 a.a. from the C-terminus of nsP4. This deletion also removes a packaging signal from RNA, so that DHs do not form nucleocapsids and will not be packaged into infectious particles even if all structural proteins are present in infected cells.

DH-delC has an additional deletion in the capsid protein gene removing a.a. 82-111. The deletion in the capsid protein gene does not change the reading frame, which is important for the translation of the downstream spike genes E3-E2-6k-E1. DH-delE carries a gene encoding the wild-type capsid protein (275 a.a.), this C gene is terminated with a stop-codon, then immediately follows 3′UTR. No envelope protein genes are present in the DH-delE helper.

A simple expression plasmid pCMV-GFP-SEAP (Figure 1g) was created from the expression vector pcDNA3.1 (ThermoFisher Cat. V79020). The gene cassette GFP-2A-SEAP was cloned between NheI and NotI sites. In this vector, a hybrid cytomegalovirus (CMV) promoter/T7 promoter controls the GFP-2A-SEAP expression. The transcription unit is terminated with a vector-supplied transcriptional termination/polyadenylation signal from the bovine growth hormone gene.

### 2.4. Mutant Variants of the Capsid Protein

Mutant variants of the capsid protein were constructed to be used in PIVs, these mutant capsid proteins being incapable of, or inefficient in encapsidating genomic RNA. Comparison of the wild-type capsid protein and constructed mutants Cmut1 and Cmut2 is shown in Figure 2.

The mutant variant Cmut1 has 28 substitutions, of which the majority removes positive charges from the RNA-binding (N-terminal) portion of the capsid protein. Because of these substitutions, Cmut1 is incapable of packaging viral genomic RNA as described in the work [12]. In addition, Cmut1 has substitutions in the nuclear localization signal (NLS) and the connecting peptide to the nuclear export signal as shown in Figure 2. Changes to the NLS and the connecting peptide are the same as in the noncytopathic capsid protein C1 described in [28]. The mutant variant Cmut2 is different in that it has 26 substitutions, all of which remove positive charges. A subset of the mutations in Cmut2 also affects NLS, thus rendering Cmut2 noncytopathic. Previously, a capsid protein with the same 26 substitutions was studied by a different group and labeled “RK-“ in [12].

### 2.5. Pseudoinfectious Viruses and a Replicable Helper

PIV differs from a replicon in that the PIV encodes all structural proteins, however its virion-forming proteins are incapable of encapsidating the genome and forming infectious particles. Two PIVs were constructed, using mutant variants of the capsid protein, one PIV with Cmut1 and another with Cmut2.

Both PIVs carry two copies of the subgenomic promoter (SP). The first copy of SP drives transcription of the subgenomic RNA from which the gene cassette GFP-2A-SEAP is translated. The second copy of SP drives transcription of smaller subgenomic RNA-encoding structural proteins.

A replicable helper used for packaging PIV genomes into infectious particles was constructed based on the defective helper DH-delE, into which complete viral replicase genes nsP1-nsP4 were cloned into their native position. Thus, the obtained helper RH-piv has a kind of similarity in the molecular design to the helper P1234/2A-C1-Cherry, which was used to package PIVs in the work [12]. The main difference between the replicable helpers is that our RH-piv has a wild-type capsid protein and P1234/2A-C1-Cherry has a noncytopathic C1 capsid protein [12]. Another difference to [12] is that RH-piv does not encode any fluorescent protein.

### 2.6. DNA-Transfections to Produce Infectious Particles

BHK-21 or HEK293FT cells were transfected using Lipofectamine 2000 (Thermo Fisher Scientific, Waltham, MA, USA, Cat. 11668019). The cells were grown in P150 dishes to 90% confluence. A plasmid mixture (20 μg total DNA) was added to 800 µL of serum-free Opti-MEM medium (ThermoFisher 11058021). Lipofectamine 2000 (20 μL) was added to another tube with 800 µL of Opti-MEM. Aliquots of DNA and transfection reagent were combined and incubated for 20 min at room temperature. The incubation medium was removed from a dish with growing cells and cells were washed with DPBS (ThermoFisher 14190136). The transfection mixture was added dropwise onto the cell monolayer, then additional 2.4 mL Opti-MEM was added. The cells were incubated with the transfection mixture for 4 h, then the liquid was replaced with 20 mL of complete growth medium, which is DMEM with 10% FBS and other additives. This time point of the addition of complete medium was considered as point zero h post-transfection (0 hpt). The culture was incubated for 2 days without medium change. The conditioned medium was collected, clarified by centrifugation, aliquoted, and stored at −80 °C.

A different procedure used to transfect cells was coprecipitation with calcium phosphate (CaPi). The author’s CaPi protocol is described with details in the previous work [29]. BHK-21 or HEK293FT cells were seeded in P150 dishes using a seeding density of 50,000 cells/cm^2^ (BHK-21) or 100,000 cells/cm^2^ (HEK293FT). Transfection was performed the following day. The total mass of DNA in the transfection mixture was 150 μg per dish. A solution of plasmids in water (in a volume of 150–500 μL) was added to a 15 mL tube, and then 0.25 M CaCl_2_ solution was added to bring total volume to 3 mL. Following that, 3 mL of 2xHeBS buffer (274 mM NaCl, 10 mM KCl, 15 mM glucose, 42 mM HEPES, 1.4 mM sodium phosphate) and pH 7.12 were added to the tube. The 2xHeBS buffer was added dropwise, with continuous mixing on the Vortex. The transfection mixture was incubated for exactly 5 min at room temperature to allow for the controlled formation of CaPi precipitates (exact timing is important). During formation of CaPi complexes, the medium was changed in the P150 dish with cells to 20 mL of DMEM base medium (without serum and other additions). The CaPi transfection mixture was promptly added to DMEM base medium in the dish and distributed over the cells. The culture was incubated for 1 h in a CO_2_ incubator, then 2.6 mL FBS were added. The next morning and no later than 18–20 h after the addition of CaPi, the medium was changed to complete growth medium with 10% FBS. This time point of the medium replacement was considered as time zero post-transfection. The culture was incubated for 2 days. The conditioned medium was collected at 48 hpt.

Transfection with the simple expression vector pCMV-GFP-SEAP was performed in P100 plates (20 μg per plate). In transfections with plasmid mixtures, the component plasmids were taken in equimolar amounts. For example, for the packaging of DNA-launched replicons (DREPs), which have 16,197 base pairs (bp), with helpers DH-delC (10,811 bp) and DH-delE (7963 bp) and using Lipofectamine, the plasmid mixture contained 9.3 μg DREP, 6.2 μg DH-delC, and 4.5 μg DH-delE. For packaging of PIVs (20,244 bp) with helper RH-piv (14,734 bp) using CaPi, the mixture was 87 μg PIV and 63 μg RH-piv.

Data presented in the figures were collected from experiments in which similar cultures were transfected in triplicates.

### 2.7. Titration of Infectious Particles and SEAP Assay

Infectious particles (replicon particles or PIVs) were titered by infecting HEK293FT cells in 6-well plates and measuring fractions of GFP+ cells using flow cytometry, as described in [29]. HEK293FT cells were seeded in 6-well plates (4 × 10^5^ cells/well) and allowed to attach for at least 4 h. Dilutions of a sample of infectious particles were prepared in complete growth medium. These dilutions were added to the wells to replace the existing medium. Upon incubation for 24 h, the cells were detached with trypsin/EDTA, resuspended again in complete medium, pelleted by centrifugation, and resuspended in 1 mL of PBS/EDTA buffer (PEB, Miltenyi Biotec Cat. 130-092-747). Fractions (%) of GFP+ cells were counted using MACS Quant 10 flow cytometer (Miltenyi Biotec). GFP was detected in channel B1 488/(530/30) nm. Channels for forward scatter (FSC) and side scatter (SSC) were used to gate out debris. The fractions of GFP+ cells were recalculated into infectious particles titers. The titers are expressed in focus-forming units (FFU).

SEAP was measured using a test with p-nitrophenylphosphate (pNPP, Sigma-Aldrich, St. Louis, MO, USA, Cat. 4876) as the substrate. Substrate solution contains 1 mg/mL pNPP and 20 mM homoarginine (Sigma Cat. H1007) dissolved in an alkaline buffer (1.0 M triethylamine, 0.5 mM MgCl_2_, pH 9.8). Samples of the conditioned medium were heated at 65 °C for 30 min to inactivate endogenous phosphatases, then centrifuged to remove debris. The samples and substrate solution were equilibrated at 37 °C before testing. The test was performed in 96-well plates. The substrate solution was dispensed by 100 µL per well. Aliquots (100 µL) of the conditioned medium were added to the wells of the first row (row A), the contents in the wells were mixed. Immediately, 100 µL-portions were transferred to the second row (B), mixed, and so on. The plate was sealed and incubated at 37 °C for 30 min. Optical densities (OD) were measured at 405 nm using plate reader ChroMate-4300 (Awareness Technology Inc., Palm City, FL, USA). Each experiment included controls. The medium collected from a culture of naïve HEK293FT cells was used for negative control (NC); this medium was also heated at 65 °C for 30 min. ODs from the NC were subtracted from readings of actual samples. Positive control: commercial recombinant SEAP protein (Invivogen, San Diego, CA, USA, Cat. rec-hseap) and in some cases alkaline phosphatase FastAP (Thermo Fisher Scientific, Waltham, MA, USA, Cat. EF0651) were used for the positive control (PC) and to build calibration curves.

To build calibration curves, the recombinant SEAP protein was dissolved to 100 ng/mL. When FastAP was used, it was dissolved to a concentration of 20 units/mL. The diluent was a heated medium (same as for the negative control). The dissolved enzymes were added to wells in row A (already containing the pNPP substrate). Other wells in a vertical row were filled with successive 1:2 dilutions of the enzymes. Authors determined a mass concentration of FastAP in commercial preparations to be 300 µg/µL.

### 2.8. Detection of Replication Competent Virus

The experiment involved four blind passages, starting with the packaged PIV vector as the initial sample. HEK293FT cells cultured in 6-well plates were infected with PIV particles or samples from previous blind passages. For each passage, 100 µL of conditioned medium per well was transferred to fresh cells for reinfection. At 48 hpi, the conditioned medium was collected, with a portion used to infect new HEK293FT cells and the remainder preserved for RT-PCR analysis.

RNA was extracted from the conditioned medium using PureLink Viral RNA/DNA Mini Kit (ThermoScientific, Cat. 12280-050) following the manufacturer’s instructions. The extracted RNA was precipitated with isopropanol in the presence of carrier tRNA, washed with 70% ethanol, and resuspended in 20 µL of nuclease-free water.

For RT-PCR, primers targeting the nsP2 gene were used: nsP2f (5′-CGTGGCTTGATAAAGGTTACC-3′) and nsP2r (5′-AAAAAACCGCACTGTTTGGGA-3′). The RT-PCR reaction was performed using the BioMaster RT-PCR-Extra 2X Kit (Biolabmix, Novosibirsk, Russia, Cat. RM06-200) in a single-tube format, following the manufacturer’s guides. The thermal cycling profile included reverse transcription at 50 °C for 50 min, initial denaturation at 93 °C for 3 min, followed by 33 cycles of 93 °C for 15 s, 59 °C for 30 s, and 68 °C for 1 min. PCR products were analyzed by gel electrophoresis.

A packaged PIV vector served as the positive control. The conditioned medium from uninfected cells was used as the negative control.

### 2.9. Setting Up Producing Cultures

Small-scale cultures were placed in P100 dishes. HEK293T cells were seeded in P100 dishes—2 × 10^6^ cells per dish—and allowed to attach for at least 4 h. The cultures were infected with replicon particles or PIV particles. Varying multiplicities of infection (MOIs) were tested in different experiments. Infectious particles were diluted in 2 mL of DPBS (ThermoScientific Cat. 14190144) containing 1% FBS. Existing media were removed, the infectious inocula were distributed in the dishes, which were incubated for 1 h with occasional shaking. Then, the inocula were discarded, and the cells were supplemented with 10 mL of complete medium. In time-course experiments, media were collected every day with a complete medium change. Collected samples were stored at −80 °C before measuring SEAP.

Large-scale producing cultures were in five-layer stacks Multilayer Cell Culture Flasks (VWR, Radnor, PA, USA, Cat, 734-3418). HEK293T cells were grown in 6-7 P150 dishes to obtain seeding amounts of the cells. The cells were trypsinized and transferred into a five-layer stack with seeding density 100,000 cells/cm^2^. In some experiments, a fixed number of 180 million cells were seeded into the stack on the day of infection. To infect the cells, PIV particles were diluted in 50 mL of DPBS + 1% FBS. A range of MOIs were used in different experiments. The medium was removed from the stack, and the infectious inoculum was poured in and distributed over the shelves. The stack was incubated for 1 h with occasional shaking. Then, the DPBS was removed, and 100 mL of complete medium added. The conditioned medium was collected at 48 h post-infection (hpi). The collected medium was clarified by centrifugation (6000 rpm 10 min) and stored before processing at −80 °C.

### 2.10. Purification of Recombinant SEAP Using IMAC

Triton X-100 was added to the conditioned medium to a final concentration of 1% and the medium was incubated for 15 min at room temperature. After that, 1/10 volume of concentrated (10X) buffer A for metal affinity chromatography (IMAC) was added. The 10X buffer A contains 500 mM sodium phosphate, pH 8.0, 1.5 M NaCl, 500 mM imidazole. Then, nickel chloride (NiCl_2_) solution was added to the desired final concentration of nickel ions in the buffered medium. The optimal nickel ion concentration was determined in preliminary experiments. The majority of purifications were performed using 5 mM Ni^2+^. After all additions, if turbidity appeared after the addition of Ni^2+^, the medium was centrifuged (6000 rpm 10 min) to remove insolubles. The supernatant was collected for further processing.

Adsorption of SEAP-10xHis on IMAC resin was performed as follows. To 45 mL of the clarified supplemented medium, 1 mL of 50% suspension of IMAC Sepharose 6 Fast Flow resin (Cytiva, Marlborough, MA, USA, Cat, 17092108) was added. The resin was pre-loaded with Ni^2+^ ions according to the manufacturer’s protocol. The tubes were then incubated overnight at +8 °C applying continuous turn-over mixing on a mixer (SB2 rotator, Stuart, UK).

The next day, the resin was transferred to an empty column PD-10 (Cytiva, Marlborough, MA, USA, Cat. 17043501). The resin was washed twice with 10 mL of wash buffer (50 mM sodium phosphate pH 8.0, 0.5 M NaCl, 50 mM imidazole). Then, the column outlet was capped, the resin was resuspended in 2 mL of elution buffer (50 mM sodium phosphate pH 8.0, 0.5 M NaCl, 250 mM imidazole), and incubated for 30 min. Upon this incubation, the elution buffer was allowed to flow out and was collected to form fraction #1. In the analogous fashion fractions, 2-4 were produced sequentially. This material was concentrated 10X using Amicon Ultra-2 Centrifugal Filter Devices (Sigma Cat. UFC201024) with MWCO 10 kDa.

Western blot with the purified protein was performed using the anti-His Tag Monoclonal Antibody (HIS.H8), HRP conjugate (Thermo Fisher Scientific, Waltham, MA, USA, Cat. MA1-21315-HRP) as a single staining antibody. To reveal immune complexes, the blotting membrane was incubated in 4-chloro-1-naphthol substrate (Merck KGaA, Darmstadt, Germany, Cat. C8302) with added (0.01%) hydrogen peroxide.

### 2.11. Statistical Processing of Results

Transfections to produce replicon particles or PIV particles were performed in triplicates. Data for SEAP purification were combined from a series of identical experiments. Data were analyzed in GraphPad Prism (GraphPad Inc., San Diego, CA, USA). Statistical significance (*p*-value) for differences between means was computed using unpaired two-tailed Student’s *t*-test. Symbols to denote *p*-values were the following: ns (*p* > 0.05); * (*p* ≤ 0.05); ** (*p* ≤ 0.01); *** (*p* ≤ 0.001). Graphs show arithmetic means and standard deviations (SDs).

## 3. Results

### 3.1. Titers and Recombinant Protein Production Using Replicon Vectors

Replicons of VEEV were created as expression vectors, and the replicons were devised to express GFP and SEAP. This study utilized two cell lines, BHK-21 and HEK293FT, selected based on published experience in alphavirus research. The BHK-21 cell line is widely used in alphavirus studies [22,30,31], primarily because it cannot suppress viral replication and supports persistent infection [32,33]. HEK293FT was chosen for its high chemical transfection efficiency, confirmed in the authors’ prior work [34]. However, it only supports the acute phase of alphavirus infection and inhibits VEEV replication in long-term experiments [35]. This resistance is because HEK293FT cells establish an antiviral state that depends on IFN-I signaling. Although this study did not aim to investigate the impact of the antiviral state on expression, it was of interest to compare vectors with varying sensitivity to the antiviral response in HEK293FT cells. To this end, two different replicon vectors were designed and tested. The literature describes mutations in nsP2 or nsP3 of VEEV that enable replicons to replicate indefinitely in IFN-I-competent cells like HEK293FT [25]. A Q739L mutation [25] was introduced into nsP2 in one replicon, while the other retained the wild-type nsP2.

Replicon particles were produced in BHK-21 and HEK293FT cells transfected with mixtures of plasmids: DNA-launched replicons (DREPs) and defective helpers (DH-delC + DH-delE). In initial experiments, BHK-21 and HEK293FT cells were transfected in identical conditions (at 90% confluence with Lipofectamine). Microscopic examination of the cultures showed similar fractions of GFP+ cells (~70%) at 48 hpt. However, titers of replicon particles accumulated at 48 hpt appeared much higher for HEK293FT cells compared to BHK-21 cells and the observed difference was particularly high for the wild-type-nsP2 replicon VEErepGS (Figure 3a).

This result was unexpected because, as was mentioned, HEK293FT can suppress the intracellular replicon, which is not the property of BHK-21. These results were confirmed multiple times. In addition, the titers were always higher for the wild-type-nsP2 replicon, compared to the nsP2 mutant, despite the fact that the Q739 substitution was expected to give the mutant an advantage of long-term replication in HEK293FT. The explanation seems to be in the relatively short-term phase of active production of packaged replicons, which is most efficient during acute infection. It is probable that wild-type replicase allows for quicker RNA replication and its replicon has higher peak intracellular concentration, which promotes packaging even in IFN-competent cells.

Then, replicon particles produced in transfection experiments were used to infect BHK-21 and HEK293FT cell cultures using one MOI, 5 FFU/cell. Almost all cells in the infected cultures were GFP+ at 48 hpi (Figure 3b). However, the intensity of GFP fluorescence was visibly higher for the wild-type replicon as compared to nsP2-mutant (Figure 3b). All replicon-infected cultures produced the secretory product SEAP (Figure 3c). And it appeared that HEK293FT produced more SEAP than BHK-21 (infected with the same replicons). It follows from this production, that the antiviral state in producing culture is not an obstacle for high-yielding TGE, even in cells like HEK293FT, which are known to clear replicons during long observation.

Comparison of the SEAP activity accumulated to 48 hpi in the replicon-infected cultures and in simple plasmid-transfected cultures (pCMV-GFP-SEAP was used) is presented in Figure 3d. The wt replicon produced much higher levels than the simple plasmid, and the nsP2-mutant replicon produced less. These results convinced us to abandon further use of the nsP2-mutant. Because HEK293FT performed better as producers, further experiments to produce the SEAP were performed in HEK293FT.

### 3.2. Pseudoinfectious Viruses as Expression Vectors

It was not possible for authors to compare the production using a recombinant monopartite-genome VEEV bearing the same gene cassette GFP-2A-SEAP, because of the regulation for handling human pathogens in the authors’ organization. However, this regulation does not prohibit creating split-genome viruses that are incapable of causing self-supported productive infection in vivo. Pseudoinfectious viruses (PIVs) are different from natural viruses, in that PIVs replicate and can give increasing titers only in cell cultures that contain a different molecule to complement the defect in PIVs packaging. A seminal paper, which for the first time disclosed alphaviral PIV [12], also describes several helpers. For the work herein, two PIVs and a replicable helper were assembled for which the molecular designs are shown in Figure 4a,b. Our replicable helper RH-piv (Figure 4b) is different in many aspects from helpers described in [12]. An exemplary difference is that our RH-piv has wild-type and cytotoxic capsid protein, resulting in the presence of cytotoxicity, which appears to be beneficial to the expression (described in the next paragraphs).

Packaging of the PIVs into infectious particles was achieved by transfecting HEK293FT cells with plasmids (MICs for PIV and RH-piv). Transfected cultures produced infectious PIV particles to high titers approaching 10^9^ FFU/mL during the first two days (Figure 5a). Then, the titers of new particles dropped, whereas the majority of cells died, most probably because of the cytopathic action of the capsid protein encoded in the helper. The two PIVs showed identical time courses for titers in the packaging experiment. Separate cultures of HEK293FT cells were transfected with PIV vectors alone (without the helper) to assess the residual ability of the PIVs to generate infectious particles. The titers of infectious particles produced by PIVs did not exceed 10^3^ FFU/mL, and by the third day, the titers had dropped to undetectable levels. We attribute this decline in titers to the fact that IFN-competent HEK293FT cells halted the replication of PIVs when they were present without the helper genome.

Since replication-competent viruses may arise through recombination between the genomes of PIV and RH-piv, potentially compromising biosafety, PIV particle preparations were analyzed for the presence of replication-competent viruses. To assess this, four blind passages were performed, starting with the initial PIV particles preparation, and ongoing replication was monitored using PCR targeting the nsP2 gene, a key component of the viral replicase. Experimental results are presented in Appendix A. Vector particles were detected in blind passages 1–3 but not in passage 4. Additionally, the amount of amplification products decreased with each successive passage. This indicates that the PIV vector and helper function as a bipartite system capable of a limited number of infectious passages in culture. However, the number of such passages is restricted. The absence of amplification in passage 4 confirms that the preparations do not contain replication-competent viruses.

Samples of the medium containing high titers of PIV particles from the packaging experiment were used to infect naïve HEK293FT cells. A set of cultures were infected using MOIs in the range of 0.25–50 FFU/cell to test if the MOI influences the levels of expression product. Indeed, the production increased with higher MOIs, achieving a plateau at 25–50 FFU/cell (Figure 5b). The recombinant protein production had a typical TGE profile, as SEAP peaked on the second day and then declined.

The majority of cells died in the producing cultures quickly after the production peak (Appendix A). The CPE in the producing culture can be explained by the replication of the RH. As compared to DH (used to package replicons), the RH RNA preserves the packaging signal that is in nsP1 [36]. Samples obtained during co-transfection packaging are expected to contain a mixture of infectious particles, with either PIV or RH. Accordingly, the producing cultures are expected to be coinfected and the mixed infection seems to be beneficial to the expression levels. As depicted in Figure 5b, SEAP production increases with the MOI, which can be attributed to the higher number of cells coinfected with PIV and RH. The two PIV variants (having Cmut1 or Cmut2) showed similar SEAP production; for this reason, Figure 5b presents data for one PIV, and only a variant with Cmut1 was used in further works.

The SEAP expressions from different constructs in this work were compared in one experiment with high MOI, 25 FFU/cell. A control culture of HEK293FT cells was transfected with pCMV-GFP-SEAP using an efficient CaPi method. The results for measuring SEAP are presented in Figure 6a,b. Again, the PIV-based expression system is much more yielding when compared to the replicon-based expression system or simple expression vector-based TGE.

### 3.3. Purification of His-Tagged SEAP from Conditioned Medium

HEK293FT cells were seeded into 5-shelf culture stacks at a density of 100,000 cells/cm^2^ and infected the following day with PIV at an MOI of 25 FFU/cell. In some experiments, the total cell count per stack on the day of infection was approximately 180 million. Two days post-infection, the conditioned medium was collected and supplemented with 1% Triton X-100 to inactivate any remaining infectious particles. The use of Triton X-100 to deactivate residual infectivity has been shown to be effective in previous studies with VEEV PIVs [12].

His-tagged SEAP was purified using IMAC. However, when the standard IMAC protocol was applied, feeding the IMAC column with serum-containing conditioned medium led to the desorption of metal ions from the resin (evidenced by the resin turning white), and the His-tagged SEAP was not captured at all. We hypothesized that substances in the chromatographic feed, which have a higher affinity for metal ions than the IMAC resin, were chelating the metal and leaching it from the resin. The presence of serum was found to be beneficial for expression, and we continued using the FBS-containing medium.

We proposed that the substances causing the Ni^2+^ ion leaching in the medium could be “neutralized” by adding a nickel salt to the medium at a defined concentration. Indeed, adding nickel chloride to a concentration of 5 mM to the chromatographic feed prevented the metal stripping. Additionally, we observed that the addition of imidazole was crucial, as serum proteins can precipitate in the presence of free heavy metal ions if imidazole concentration is insufficient. Increased turbidity and precipitation of insoluble materials occurred when Ni^2+^ was added to the complete medium without the prior addition of imidazole. Results from one optimization experiment are shown in Figure 7a (a photograph of the full gel is in Appendix A). In this experiment, 10 mL samples of the conditioned medium containing the expressed SEAP-10xHis were supplemented with phosphate buffer, imidazole (to 50 mM), and nickel chloride at varying Ni^2+^ concentrations. SEAP was then adsorbed onto 0.5 mL of 50% resin slurry. After extensive washes, the product was eluted with 0.5 mL of elution buffer (250 mM imidazole). The results indicated that, when using DMEM + 10% FBS as the expression medium, the optimal Ni^2+^ concentration is 5 mM.

In the final optimized protocol, the concentrations of imidazole and Ni^2+^ are 50 mM and 5 mM, respectively. These conditions effectively prevent turbidity and serum protein precipitation.

We applied the optimized conditions to perform IMAC purification from approximately 100 mL of culture medium. With this volume, the tagged product was adsorbed onto 1 mL of 50% resin slurry (pre-loaded with Ni^2+^), and the product was eluted in 2 mL fractions. The results of a representative purification are shown in Figure 7b,c. Western blotting was performed using an anti-His-tag antibody and chloronaphtol staining as described in the Materials and Methods. The results showed intense staining of the purified product as shown in Figure 7c (a photograph of the full membrane is in Appendix A). Our SEAP-10xHis has 521 amino acids and a calculated mass of 57 kDa. However, the protein contains two N-linked glycosylation sites, each adding 2–2.5 kDa, as well as numerous serine (31 Ser) and threonine (33 Thr) residues, which are potential sites for O-glycosylation. Using the DeepO-GlcNAc tool [37] to predict O-glycosylation sites, 14 potential O-glycosylation sites were identified, with each glycan adding 0.6–1 kDa. Therefore, the observed SDS-PAGE mass of slightly under 75 kDa for SEAP-10xHis is consistent with the protein’s expected size.

Protein concentration was measured using the Bradford reagent, showing a total of 11 mg of purified product. Given that 180 million cells were infected in this experiment and considering that PIV infection is cytopathic and halts culture expansion, the specific productivity was calculated to be 29 pg/cell/day. As with any TGE experiment, this specific productivity value represents an average over time, with the rate of productivity potentially varying at different stages. Nonetheless, this value highlights the high production capacity of the VEEV PIV-based expression system described.

## 4. Discussion

Industrial bioprocesses using stably transformed cell clones can show volumetric productivity at the gram per liter (g/L) scale [38]. However, such high yields have been reported for the exhaustively optimized industrial bioprocesses and not for all expressed proteins (high yields are more common for recombinant monoclonal antibodies). It was reported that the high volumetric productivity in industry is mainly because of the ability to grow producing cultures at very high cell density in bioreactors, and not because the industrial producers have per-cell productivity many folds over what is typically obtained in TGE protocols [39]. It appears that the specific productivity (per cell/day) has a natural limit that did not profoundly change (i.e., ~2-fold) in the past decades even with developments in optimizing media compositions and cell metabolic engineering [39]. Published values of the specific productivity typically do not exceed 20–50 pg/cell/day and few recent processes achieve 90 pg/cell/day [40,41,42,43,44,45,46]. A systematic study of productivity limits in HEK293 cells showed the maximum production capacity for a recombinant protein with cytoplasmic localization at 5.0% of the total protein, but for a secretory protein with an endoplasmic reticulum export signal, the limit was only 0.38%, and for protein with a mitochondria-targeting signal, the limit was only 1.6% [47]. Assuming average total protein in one HEK293 cell to be 400 pg [48,49], a mass of a secretory product undergoing processing at any moment is a mere 1.5 pg/cell. Thus, specific productivity of 20–50 pg/cell/day means that folding and glycosylation complete in just 40 min–2 h in these systems. Higher sustained expression leads to cell death [47].

Using the same industrial approaches is challenging in academic settings without access to bioreactors, as cultivating high-density producing cultures and maintaining viability are impossible. The remaining option to increase product amounts is to simply scale up producing cultures. However, scaling up transfections is also challenging [50,51,52,53].

In this study, we present a TGE technique characterized by straightforward scalability. Our protein expression method requires only minimal amounts of plasmid DNA (easily producible in a laboratory) and transfection reagents. These characteristics result from the use of a different type of expression vector—autonomously replicating RNAs—and the adoption of infection rather than transfection to scale up production cultures. The autonomously replicating RNAs were constructed from the genome of the RNA-containing virus VEEV, a species belonging to the genus alphavirus. Alphaviral vectors have been used for recombinant expression, mainly in the forms of recombinant replicable viruses or replicons [31,54,55,56], whereas PIVs are new vectors in the field. Alphaviruses as vectors have attractive characteristics, such as the ability to infect a broad range of cell types and a very efficient process of generating viral progeny. Alphavirus propagation generates a high-titered virus, above 10^9^ infectious particles per milliliter [15]. Thus, infected cultures provide virus preparations sufficient to infect much larger cultures (e.g., for passage P2 at MOI = 1, the infected culture will be 1000-fold larger than for the starting passage P1).

The use of replicable viral vectors may present biological risks [23]. In the authors’ country, regulations for handling microorganisms that can infect humans require laboratories to obtain specific licenses in addition to standard BSL certification. Consequently, the authors were unable to use monopartite-genome recombinant VEEV for expression, despite their institute having BSL-2 facilities. However, these regulations do not apply to the split-genome viral vectors described herein. The parental virus used in this study, the VEEV vaccine strain TC-83, is classified as a Risk Group 2 (RG-2) pathogen, allowing its handling under BSL-2 containment and work practices, which are standard in academic research settings. Although no replication-competent virus was detected in our preparations, even if a TC-83-like virus once emerges during the experiments, the resulting samples would still be compliant with BSL-2 biosafety requirements.

This study introduces a system for efficient TGE in academic labs. TGE is well-suited for research due to its speed (days to weeks vs. months for stable lines), flexibility (one cell line, standardized protocol for various proteins), and ability to produce small to moderate protein amounts without requiring bioreactors. While transfection-based TGE is used in the industry for early-stage development, it is less suitable for large-scale biomanufacturing due to its short-lived expression, high transfection cost, and lower productivity compared to stable cell lines, which, despite their longer development time, are optimized for high-yield production and can surpass TGE in efficiency.

Based on the results of this study, how feasible is it to introduce this system into the industry? Our vector is classified as BSL-2, making it suitable for academic research but potentially increasing commercialization costs for industrial producers, who typically prefer more cost-effective BSL-1 systems. Transitioning to higher biosafety levels (BSL-2 and above) is expected to raise costs, facility requirements, and regulatory complexity, which could make even efficient split-genome VEEV vectors less appealing for industrial commercialization.

Figure 8 provides a schematic representation of the expression system utilizing the two-component genome alphavirus PIV vector.

This study compared two types of autonomously replicating RNA vectors—alphavirus replicons and pseudoinfectious viruses (PIVs)—in terms of expression levels. A replicon is a viral genome fragment that does not encode structural proteins (i.e., proteins forming virions) but retains the UTRs and genes necessary for intracellular replication. In contrast, PIVs encode all structural proteins, but these proteins are modified to prevent genome encapsidation, thereby blocking the production of infectious particles in cells harboring PIV. Neither replicons nor PIVs can generate increasing titers in naïve cells. Although neither vector causes a productive infection in vivo, under in vitro conditions, the packaging defect can be compensated by supplying unmodified structural proteins in trans (i.e., from a separate helper construct). In the presence of a helper, both replicons and PIVs produce infectious particles at the high titers characteristic of the alphavirus genus.

Natural alphaviruses, when replicating in cell cultures, induce a pronounced cytopathic effect (CPE), leading to the rapid death of infected cells within 24 h post-infection [25]. Studies on the molecular mechanisms underlying CPE development have revealed that alphavirus replication effectively suppresses the transcription and translation of cellular mRNAs, a phenomenon known as transcriptional and translational shutoff [57,58,59]. However, the viral effectors responsible for the shutoff differ between the two primary geographic groups of alphaviruses: the Old World Alphaviruses (OWA) and the New World Alphaviruses (NWA). In NWA species, including VEEV, the capsid protein serves as the primary effector, while in OWA species, such as Sindbis virus and Semliki Forest virus, the non-structural protein nsP2 is the key mediator [59]. The capsid protein of NWA species blocks the transport of large molecules through nuclear pores, halting the transcription and translation of cellular mRNAs while allowing viral RNA processes to continue [59].

In earlier studies, the alphavirus-induced CPE was considered an important barrier to using alphaviruses as expression vectors [55]. It was considered desirable to extend the expression period before host cells die, and research focused on identifying attenuating mutations in nsP2 (for OWA species) that reduce replication rates and enable noncytopathic replication [58]. Studies on VEEV led to the development of mutant capsid proteins that can still package viral RNA but do not trigger transcriptional and translational shutoff. These noncytopathic capsid proteins contain substitutions in the nuclear localization signal (NLS) and a short peptide between the NLS and the nuclear export signal (NES) [12,28,60]. As a result, VEEV replicons lacking the capsid protein, as well as engineered VEEV-derived viruses (including PIVs) with the noncytopathic capsid protein, can replicate in permissive cells without causing noticeable CPE [12,28,60].

When using alphaviruses as vectors for TGE, the authors realized that the primary goal is not maintaining the viability of the producing cultures but rather achieving high yields of the target protein. The presence of dying cultures at the end of the experiment does not conflict with the TGE approach. In fact, we observed that the development of CPE is associated with efficient expression. Although CPE was evident at the end of the production phase in this study, the duration of 2 days was adequate to obtain milligrams of the desired product.

VEEV replicons can establish chronic infections (persistence) in cells that are deficient in producing IFN-I or responding to IFN-I [61]. Examples of such cells include BHK-21 and Vero, which do not establish an antiviral state. However, in IFN-competent cells (e.g., HEK293), infection with replicons only supports the acute phase of infection, which lasts for a few days. After this period, the antiviral state eliminates replicons (with wild-type replicase) from nearly all cells in the culture [61]. Studies have identified mutations in the replicase proteins nsP2 and nsP3 that reduce replication rates, allowing the mutant replicon to persist indefinitely in IFN-competent cells [25]. In this study, one replicon carried a mutation in nsP2 (Q739L). However, this mutant replicon did not produce higher levels of the recombinant protein compared to the wild-type version in IFN-competent cells; in fact, the levels were lower. The conclusion is that for an expression vector, the ability to persist does not provide any advantage during TGE.

Two cell lines were tested for production capacity during TGE in this study. BHK-21 cells were included because they are frequently used in alphavirus research, and a substantial body of studies has been published on the replication of engineered alphaviruses or replicons in BHK-21. This allows for comparisons between PIV and other alphaviral vectors. HEK293FT cells were selected based on the authors’ previous experience with this cell line, particularly due to its high transfection efficiency.

In the experiments described in the article, we found that HEK293FT cells have another advantage: they consistently exhibit higher expression levels, although the reasons for this superior production remain unclear. Notably, BHK-21 and HEK293FT differ in their ability to produce IFN-I and establish an antiviral state. In this context, the higher expression observed in IFN-competent HEK293FT cells is counterintuitive. This study did not specifically aim to compare cell lines with different IFN sensitivities. However, when working with viral vectors, IFN effects must generally be taken into account, as the antiviral response can decrease replication and production levels.

For future studies, it would be interesting to explore the possibility of inhibiting the IFN-induced antiviral response in high-producing HEK293FT cells through an approach that does not involve the alphavirus capsid protein or, potentially, in combination with the capsid protein in the system, as this might further enhance production levels.

In the described TGE system, the role of the capsid protein is multifaceted. First, the VEEV capsid protein suppresses the antiviral response, thereby promoting replication in IFN-competent cells, which should enhance immediate production. Second, intracellular capsid protein leads to transcriptional and translational shutoff and a rapid development of CPE, which may limit the duration of the effective production phase.

In our experiments, PIV produced significantly higher levels of recombinant protein compared to replicons. During packaging experiments, cultures transfected with replicons and defective helpers, or PIV and replicable helpers, developed CPE within 2 days. However, cultures infected with replicon particles did not show CPE until the end of the experiment. In contrast, PIV-infected cultures developed CPE throughout the production phase.

The PIV genome encodes a mutant version of the capsid protein with numerous substitutions in the N-terminal region [12,28,60]. This mutant capsid protein is nearly incapable of encapsidating viral RNA. Additionally, some substitutions disrupt the nuclear localization signal (NLS), preventing the mutant capsid protein from inducing transcriptional and translational shutoff. It was also reported that mutating the NLS and the NLS-NES connecting peptide, as performed in the Cmut1 variant, prevents the accumulation of compensatory mutations that result in increased titers in the absence of a helper [12]. Since the PIV itself is not cytopathic, the CPE observed in producing cultures is likely due to coinfection with a helper virus, which encodes the wild-type capsid protein. More recombinant protein was produced with the PIV + replicable helper pair than with the replicon + defective helper pair, despite the presence of CPE in the former. We hypothesize that the higher expression and CPE result from the same mechanism of transcriptional and translational shutoff. It is likely that in this two-component virus system (PIV + RH), cellular translation is redirected to produce only viral proteins encoded in the viral RNAs.

The data presented here demonstrate how alphaviral PIV expression vectors enable rapid and high-yield expression of recombinant proteins. The authors were unable to find other published studies that specifically use alphaviral PIVs as expression system vectors, so we cannot directly compare our results with similar work.

Another key feature of the described expression system is the method for purifying His-tagged recombinant proteins using IMAC from a serum-containing culture medium. IMAC is widely used for the capture stage of protein purification due to its simplicity and cost-effectiveness. The described modification enables the use of IMAC with standard NTA resins to capture the product, even when the chromatographic feed contains substances that could leach metal ions from the resin.

## 5. Conclusions

The use of pseudoinfectious alphavirus vectors (PIV) for transient gene expression (TGE) presents several advantages over conventional plasmid-based systems. A key benefit is that this system facilitates protein production without the requirement for large-scale transfections or costly infrastructure like bioreactors. Additionally, recombinant protein yields in the milligram range can be achieved through static cultures cultivated in multi-shelf cell culture stacks. This approach is especially well-suited for transient expression in academic research environments.

## Figures and Tables

**Figure 1 biomolecules-15-00274-f001:**
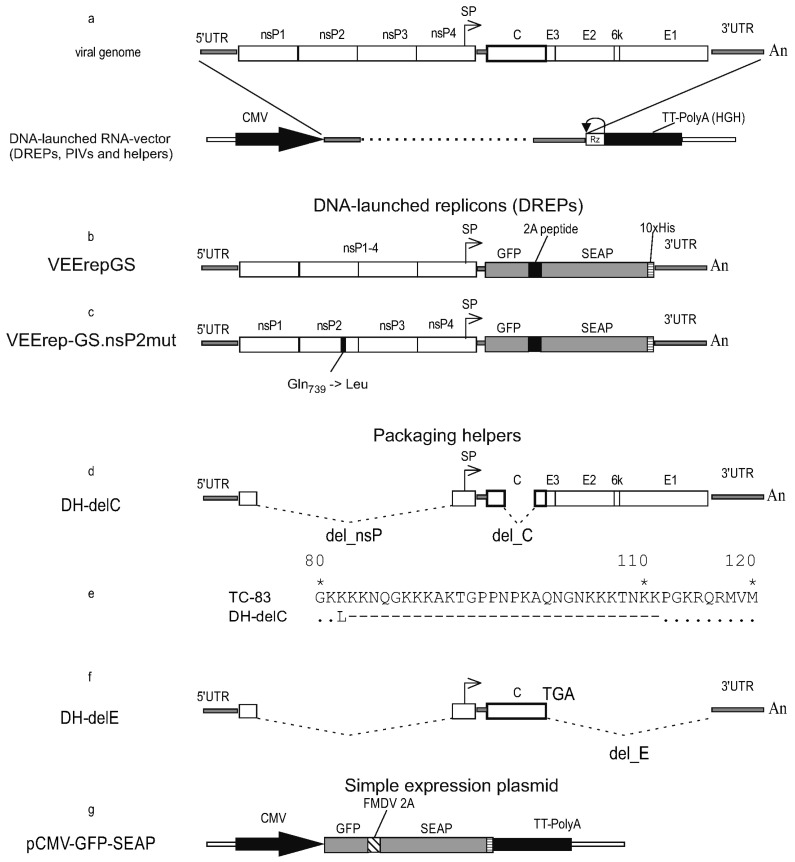
Schematic of creating DNA-launched molecular infectious clones for RNA-genomes. Molecular designs of alphaviral replicons and packaging helpers. (**a**) The genome organization of Venezuelan equine encephalitis virus (VEEV) and cloning it into a plasmid (molecular infectious clone) suitable to infect cells with autonomously replicating RNA using DNA-transfection. (**b**) The VEErepGS replicon lacks genes encoding the viral structural proteins and has a heterologous gene cassette cloned under the control of the subgenomic promoter. (**c**) The VEErepGS.nsP2mut replicon is different from VEErepGS only in that VEErepGS.nsP2mut has mutation Q739L in the nsP2 gene. (**d**) Defective helper DH-delC has a large deletion in the genes for non-structural proteins (labeled del_nsP). The helper is incapable of autonomous replication. A capsid protein gene also has a deletion, which removes amino acid residues (a.a.) 82-111. Genes for spike proteins are without modifications. (**e**) Comparison of a deletion-containing capsid protein (in DH-delC) and wild-type capsid protein from VEEV (TC-83). Matching residues are shown as dots, missing residues as dashes. Stars indicate the numbered a.a. residues. Non-homologous Leu residue is a product of translation of the restriction site AAGCTT (Lys-Leu). (**f**) Defective helper DH-delE has the del_nsP deletion and all genes for spike proteins deleted. A gene for capsid protein is wild-type (TC-83) and is terminated with an engineered TGA codon. (**g**) A simple expression vector, which is a plasmid in which the GFP-2A-SEAP gene cassette is placed downstream of the CMV promoter. Symbols: Viral genes are shown as open boxes. The heterologous gene cassette GFP-2A-SEAP is shown as gray boxes (GFP, SEAP) and a filled (2A-peptide) box. Black arrow on line with the genome is the cytomegalovirus (CMV) promoter. Antigenomic ribozyme of hepatitis D virus (Rz) and transcriptional termination/polyadenylation signal (TT-PolyA) are labeled. Arrow above diagrams show subgenomic promoter. Inscriptions: nsP (nonstructural proteins), C-E3-E2-6k-E1 (structural proteins), 5′UTR and 3′UTR (untranslated regions), An (oligo-adenine stretch), SP (subgenomic promoter) GFP (green fluorescent protein), SEAP (human placental secretory alkaline phosphatase), FMDV 2A (foot-and-mouth disease virus autoprotease 2A), 10xHis (histidine tag), del_nsP (deletion in nonstructural genes), delC (deletion in the capsid gene), delE (deletion of spike protein genes), Q739L (Gln to Leu substitution in nsP2).

**Figure 2 biomolecules-15-00274-f002:**
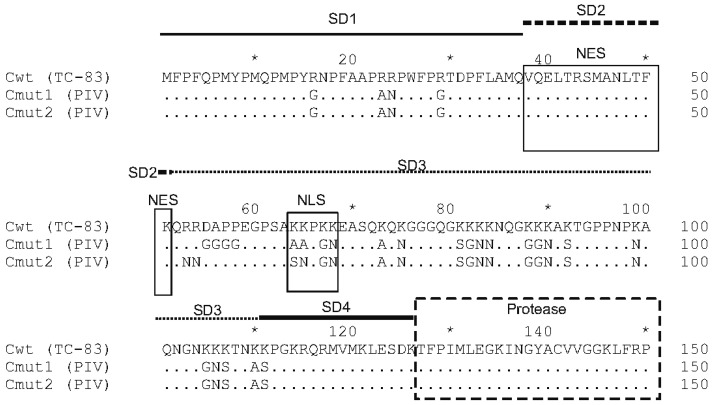
Comparison of the wild-type capsid protein (from VEEV TC-83) and two mutant variants incapable of RNA packaging. Only the first 150 a.a. of the capsid protein are shown. TC-83, wild type. Cmut1, mutant variant having substitutions in the N-terminal part (RNA-binding domain), which reduces the RNA binding. Positively charged residues are replaced with neutral-charge residues. Cmut2, mutant variant with a different set of substitutions. Subdomains in the N-terminal part of the protein are shown (SD1-SD4) in accordance with [27]. Nuclear export signal (NES), nuclear localization signal (NLS) and the connecting peptide are involved in the cytopathic action of the wild-type capsid protein (labeled). Identical residues are depicted as dots. Stars indicate the a.a. numbered at every 10 residues. Numbering of residues is given above Cwt line.

**Figure 3 biomolecules-15-00274-f003:**
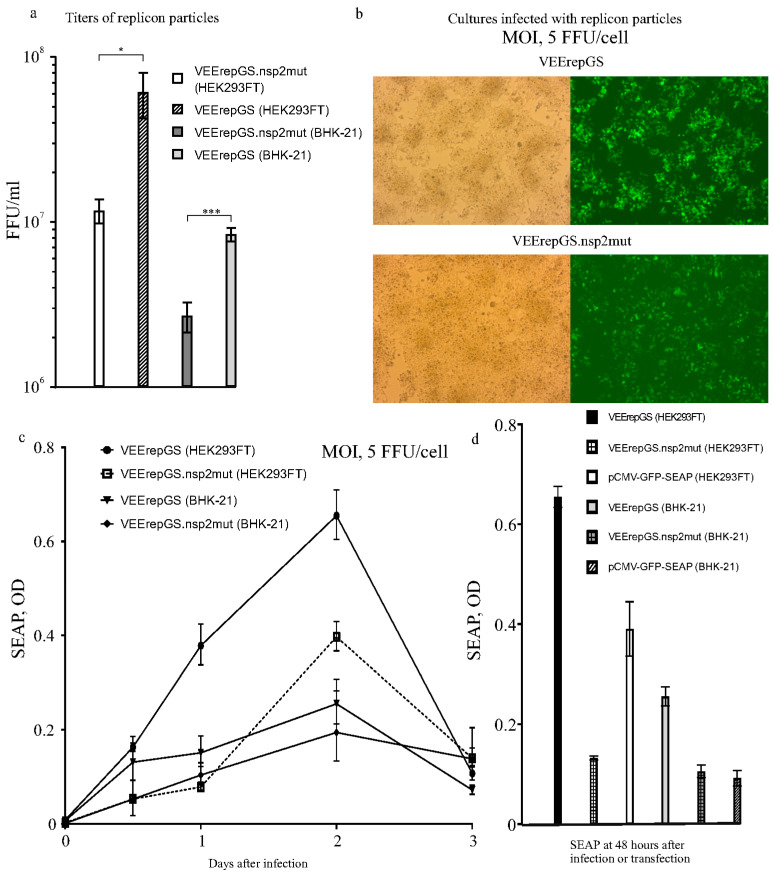
Titers of infectious particles (packaged replicons), photographs of producing cultures and SEAP levels obtained with the replicon vectors and a simple-expression plasmid. (**a**) Packaging of replicons into replicon particles was achieved by transfecting cell cultures with DREPs and helpers. The titers are shown for wild-type-nsP2 and mutant nsP2 (Q739L) replicons. Statistical significance: * (*p* < 0.05), *** (*p* < 0.001). (**b**) Pictures of HEK293FT cells infected with replicons VEErepGS (wild-type replicase) or VEErepGS.nsp2mut (nsP2 mutant). Fluorescence of GFP+ cells was photographed using the same exposure time. (**c**) Time course of the SEAP production in cultures infected with replicon particles using high MOI. HEK293FT cells produced more SEAP than BHK-21, and the wild-type replicon produced more than the mutant replicon. *Y*-axis, optical densities (ODs) obtained in the SEAP test. *X*-axis, time after infection. (**d**) Comparison of replicon vector-infected cultures, and cultures transfected with an expression plasmid, for the levels of produced recombinant protein. *Y*-axis, results of the SEAP test. *X*-axis, time after infection or transfection. Shown are means and SDs.

**Figure 4 biomolecules-15-00274-f004:**
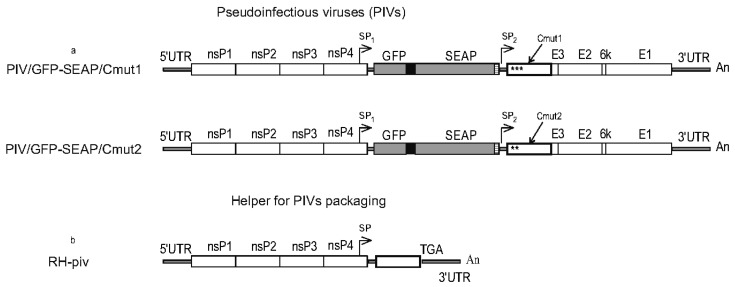
Overview of pseudoinfectious viruses and the replicable packaging helper. (**a**) A PIV is based on the VEEV genome. The capsid protein in PIV is a mutant incapable of encapsidating genomic RNA. Clusters of mutations in the N-terminal part of the capsid protein, specifically Cmut1 and Cmut2, are denoted with stars *** and **, respectively. A second copy of subgenomic promoter (SP) is present in the PIV genome. Gene cassette GFP-2A-SEAP is cloned under the control of SP1 and structural protein genes (including mutant C) are under the control of SP2. The two PIVs are different depending on the version of the mutant capsid protein. (**b**) Replicable helper RH-piv has complete genes for replicase nsP1-nsP4 as well as cis-acting sequences 5′UTR and 3′UTR. Only the capsid protein gene (wild-type) is present in the helper in the structural genes region. Symbols are the same as in Figure 1.

**Figure 5 biomolecules-15-00274-f005:**
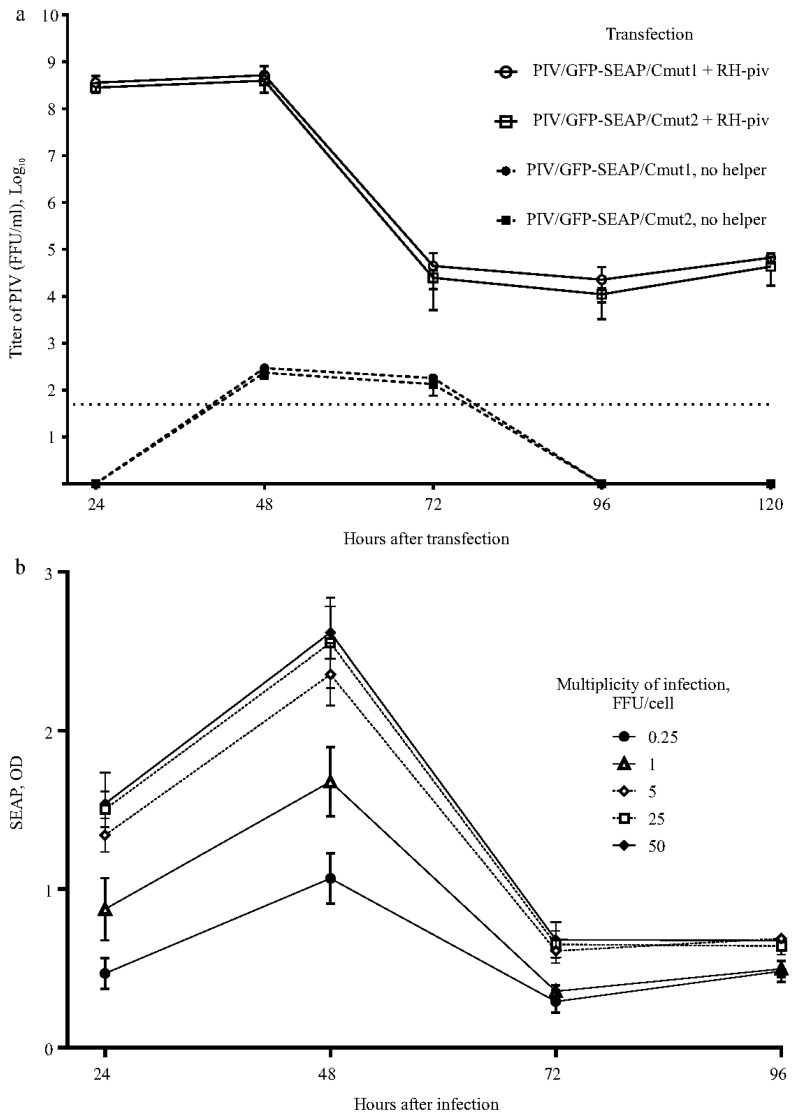
Titers of infectious particles (pseudoinfectious viruses, PIVs). Levels of the recombinant protein in cells cultures infected with PIV. (**a**) Packaging cultures (PIV + RH-piv helper) are expected to generate a mixture of two types of infectious particles: one containing the PIV genome and the other containing the helper genome. The method used to quantify infectious particles is based on counting GFP+ cells, which specifically measures the PIV-containing particles. Separate cultures were transfected with only PIV vectors (no helper) to measure the residual capacity of the PIVs to produce infectious particles. The horizontal dotted line represents the detection limit. (**b**) Production of the recombinant protein SEAP in PIV expression vector-infected cells cultures. Triplicate sets of cultures were infected with the PIV using different MOIs given in the figure legend. *Y*-axis, optical density in the SEAP test. *X*-axis, time after infection. Graphs show means and SDs.

**Figure 6 biomolecules-15-00274-f006:**
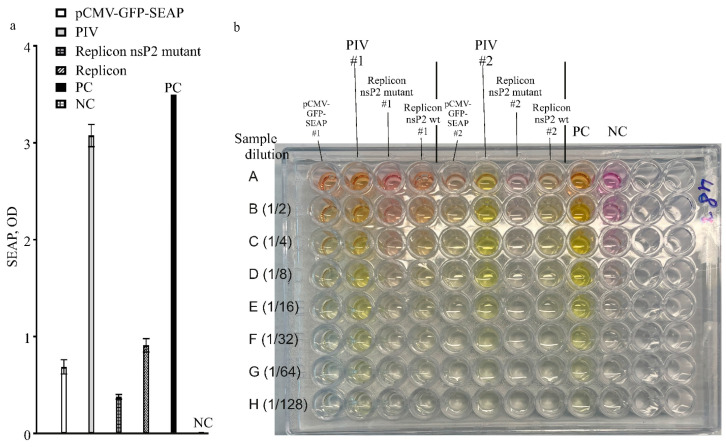
Production of the recombinant protein in cell cultures bearing expression RNA vectors (replicons, PIV) or a simple expression plasmid. (**a**) Expression analysis by the levels of SEAP produced by expression vectors: PIV, replicons, and a simple expression plasmid. Alphavirus genome-based constructs are PIV (pseudoinfectious virus with mutant capsid protein variant Cmut1), VEErepGS (replicon with wild-type replicase), VEErepGS.nsp2mut (replicon, nsP2-mutant). Simple expression plasmid is pCMV-GFP-SEAP. Producing cultures were obtained by particles infection (MOI 25 FFU/cell for the PIV and replicons), or DNA-transfection using CaPi for the simple vector. Expression media were collected on day 2 after infection or transfection. PC, positive control (commercially available SEAP, 1 unit in the same reaction conditions as other samples). NC, negative control (medium from naïve cells). *Y*-axis, optical density in the SEAP test. Plotted are means and SDs. (**b**) Photograph of one reaction plate from SEAP test used to measure the levels shown in panel (**a**). Two replicates were assayed on the presented plate (# is the replicate number). Aliquots of samples of expression media were loaded in row A of the test plate. Other rows contain dilutions of contents in row A as indicated at the left of the photograph.

**Figure 7 biomolecules-15-00274-f007:**
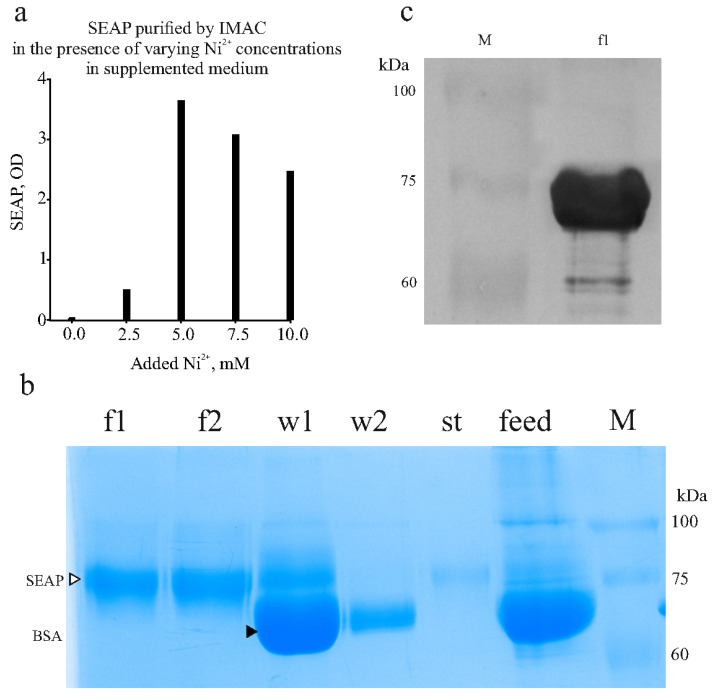
Finding optimal nickel ion concentration and SEAP purification. (**a**) Finding optimal concentration of Ni^2+^ to add to supplemented chromatographic feed to prevent Ni^2+^ leaching from IMAC resin. Activity of SEAP in eluates is shown. (**b**) SDS-PAGE with samples collected during SEAP purification from culture medium. F1,f2, fractions eluted in 250 mM imidazole. W1,w2, material washed from IMAC resin (two successive washes) with 50 mM imidazole. St, material collected upon stripping resin with 50 mM EDTA. Feed, conditioned medium. M, molecular mass marker. Bands of SEAP and BSA labeled with right-pointing triangles. (**c**) Western blot with anti-His-tag antibody. F1, sample from panel (**a**). Bands of molecular mass marker (M) remain visible on the membrane.

**Figure 8 biomolecules-15-00274-f008:**
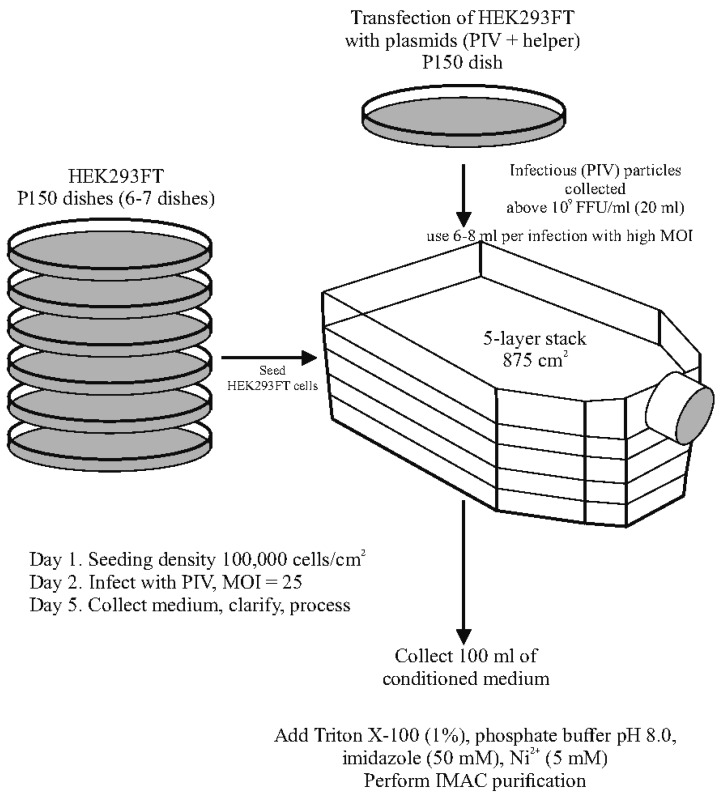
Approach for recombinant protein production using pseudoinfectious viral vectors.

## Data Availability

Data are contained within the article or Appendix A.

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
