# Peer review of "Application of Pseudoinfectious Viruses in Transient Gene Expression in Mammalian Cells: Combining Efficient Expression with Regulatory Compliance"

_biomolecules, 2025, doi:10.3390/biom15020274_

Round 1
Reviewer 1 Report
Comments and Suggestions for Authors
The study by Zauatbayeva and colleagues, from Alexandr V. Shustov Lab, aims to optimize the use of the VEEV alphavirus as a recombinant gene expression system (for protein purification). The pseudoinfectious virus (PIV) strategy was proposed, where the structural genes (apart from the packaging-competent capsid protein) are preserved in the VEEV genome, and the complementing function of C is provided in trans by replication-competent helper construct. The term “pseudoinfectious virus” is relatively new (although it had been previously used for recombinant dengue virus-based system); it resembles to some extent the production of viral replicon particles (VPRs) as both are capable of only a single round of infection and do encapsidate replicon RNA if such a function is provided in trans. In fact, the study compared the two types of autonomously replicating RNA vectors—alphavirus replicons and PIVs in terms of expression levels, with the latter outperforming VPRs in the recombinant protein yield. This strategy is promoted by the authors as a better alternative to standard plasmid-provided transgene expression. The system is dedicated to mammalian cell expression, and such an information should be included in the manuscript’s title, as other eukaryotic systems are not included in the comparison (like: Application of pseudoinfectious viruses in transient gene expression in mammalian cells: combining efficient expression with regulatory compliance).
The benefit from the replicon system is increased replicating RNA levels which drive higher expression. As regards the costs and homogeneity of production, the authors rightfully indicate the advantage to plasmid transfection. However, they do not tackle the option of stable cell lines derived from retrovirus/lentivirus vectors. Only cell lines produced by an awkward long-lasting procedure using methotrexate or methionine sulfoximine selection are mentioned. Therefore, we cannot predict the differences in yield when easy to scale-up, transgene-integrating stable cell lines would be used, maybe this option should be better described in the Introduction.
Apart from the vector construction, the authors also describe improved adsorption of recombinant proteins from the obtained crude material (cell culture supernatant+Triton X-100) in the presence of low nickel ion concentration, and such a modification of the protocol is convincingly explained.
Generally, the procedure of amplicon vector production proposed by the authors seems to be complicated and it takes several reading sessions to understand well the concept of PIV. I suggest clear description of the two amplicon types in the Introduction (line 144). Then, even if the obtained protein yield from PIV is attractive, I have two major concerns. Firstly, the mechanism by which PIV outperforms “traditional” amplicon was only suggested, but not explored, and comparing transgene RNA levels should be feasible within the revision time. Is SP promoter more active in PIV?
Another important feature of alphaherpesvirus-based replicons, is their biosafety, and this issue should be also, in my opinion, experimentally addressed and described to convince that the PIV system is practical for general use in laboratories. Using BSL-2 conditions would increase the costs of production. The strategy of PIV with RH-piv helper seems to increase the hazard of replication-competent virus (RCV) generation (as the authors admit, during vector production, a mix of PIV and RH-piv particles is expected). The authors used the VEEV vaccine strain TC-83 with reduced pathogenicity; however, to propose BSL-1 conditions to this system, the authors should follow the existing regulations for amplicon systems, providing information on formation of RCV during PIV production (tested by appropriate assays).
The experiment described in Figure 5 should, therefore, have a negative control, without the RH-piv construct, to assess the ability of self-transmission of the VEEV genome.
Another major remark regards the conclusion on IFN-dependance of expression. In my opinion, such a conclusion cannot be drawn when comparing two cell lines originating from distinct species, as other factors, like, e.g., adaptation to restriction factors may also play a role. If the authors wish to sustain this conclusion, they should compare HEK293FT cells with human IFN-deficient cells. Otherwise, I would recommend making only an “of note” remark during discussion, that the tested cell lines differ in their responsiveness to IFNs.
Minor questions and comments:
- The manuscript is generally well-written, the editorial errors are sporadic, but some expressions seem to be awkward, like “spent medium”. Please use Italics for Latin.
- ORFs for GFP and SEAP were joined in a contiguous ORF through the 2A-peptide. What was the efficiency of GFP and SEAP separation by the 2A peptide during translation in both tested cell lines, and can it contribute to the active SEAP yield? Is it possible that the WB in Figure 7C is presenting in fact a double band (with and without GFP?), thet would be visible when less material is analyzed? could anti-GFP blot confirm the efficiency of separation? This is a minor remark, because, of course, the GFP-2A-SEAP construct was used here only as a model.
- The Western blotting in Figure 7C/Figure S5 seem not to include negative control, e.g. by comparing the lysates used for purification, this would augment the conclusion on band specificity.
Author Response
- Reviewer 1 wrote: The system is dedicated to mammalian cells… should be included in the manuscript’s title…
Authors’ response: The suggested changes were made to the Title.
- Reviewer 1 wrote: The term "pseudoinfectious virus" is relatively new in the field.
Authors’ response: Yes, this could potentially cause initial confusion for the reader. To clarify the concept of pseudoinfectious alphaviruses and distinguish them from common replicons, we revised the paragraph in the Introduction (lines 126-147), Results (lines 558-560) and Discussion (lines 789-800).
- Reviewer 1 wrote: The authors indicate… (that compared to) plasmid transfection… (there is an) option of stable cell lines derived from retrovirus/lentivirus vectors. …predict the differences in yield when easy to scale-up, transgene-integrating stable cell lines would be used… described in the Introduction.
Authors’ response: For this comment, we offer the following rebuttal: The system we developed and described in the article, based on alphavirus vectors, is designed for transient expression and eliminates the need for any selection steps to obtain productive cultures. We do not perform any amplification of high-yielding clones to increase the volume of producing cultures. Due to the transient nature of expression, we compare our system with the most commonly used transient expression systems, which rely on transfection with conventional plasmid vectors, but do not involve clone selection or amplification of the transfected culture or selected clones to generate large volumes of producing cultures.
In this comment, Reviewer 1 suggests an alternative to our approach (transient expression) by returning to stable expression. However, the proposed method using retrovirus/lentivirus vectors for generating producing cultures still requires clone selection after lentiviral transduction and subsequent amplification of the selected clones to generate sufficient cell quantities. The suggested approach is one of the stable expression methods. Since stable expression falls outside the scope of our article, we chose not to include a comparison of production levels using our method versus stable cell lines, including those obtained via lentiviral transduction, in the Discussion.
In fact, studies comparing recombinant protein production levels using viral vectors of these types—alphavirus vectors and lentivirus vectors—have already been published:
- "Evaluation of recombinant alphaviruses as vectors in gene therapy" by JJ Wahlfors et al. (Gene Ther. 2000 Mar;7(6):472-80) directly compared recombinant expression levels and temporal dynamics between an Alphavirus vector and a retroviral vector. The data, presented in Figure 4, show that the Alphavirus (Sindbis virus) vector significantly outperforms other types (retrovirus and adenovirus) in expression levels during the transient expression phase (up to 75 hours) before cell die-off begins.
- "Gene transfer into neurons from hippocampal slices: comparison of recombinant Semliki Forest Virus, adenovirus, adeno-associated virus, lentivirus, and measles virus" by MU Ehrengruber et al. (Mol Cell Neurosci. 2001 May;17(5):855-71) compared GFP expression in native cells using Alphavirus and lentivirus vectors and similarly concluded that under transient production conditions, Alphavirus provides both faster dynamics and higher production levels.
- Reviewer 1 wrote: “easy to scale-up, transgene-integrating stable cell lines would be used… described in the Introduction”.
Authors’ response: In the Introduction section, we added a paragraph mentioning that other types of expression vectors are being developed in the industry, such as lentiviral vectors capable of genomic integration, as well as similar vectors for transient expression using lentivirus vectors with defective integrase. The added text can be found in lines 66-73.
- Reviewer 1 wrote: It takes several reading sessions to understand well the concept of PIV. I suggest clear description of the two amplicon types in the Introduction (line 144).
Authors’ response: This paragraph in the Introduction (starting from line 144 in the first version of the manuscript) was revised to provide a clearer understanding of the types of vectors developed and the main results. Improved description of the concept of PIV is in lines 126-147.
- Reviewer 1 wrote: … Is SP promoter more active in PIV?
Authors’ response: In this study, the authors did not measure the levels of intracellular subgenomic RNAs synthesized in cells infected with replicons or PIVs. This is because the practical goal of the work was to develop a convenient expression system that meets the requirements for easy scalability of producing cultures without the use of large-scale transfection. At this stage, the study did not aim to investigate the intracellular mechanisms driving higher production from PIV vectors, though such studies are planned for the future. Instead, we used a direct assay to measure the SEAP product to select the best vector. The mechanisms determining higher production from PIV vectors will be explored in future work, and this will be the subject of a future publication.
However, based on previously published work by other authors, we can formulate some preliminary expectations in response to the reviewer's question. The higher levels of marker protein production from PIV vectors compared to replicons cannot be explained by the assumption that the subgenomic promoter directing the GFP-SEAP gene cassette in the replicon is more active than the subgenomic promoter driving the same gene cassette in PIVs. This is because, in replicons, the subgenomic (SG) promoter is the only promoter driving the expression of a single subgenomic RNA, leading to a focused utilization of the SG with the viral RNA-dependent RNA polymerase (RdRp). In PIV systems with two SG promoters, these promoters share the same pool of RdRp, resulting in competition between them.
Furthermore, it has been published that in alphaviruses with double subgenomic promoters (SG1 and SG2), the second copy (designated SG2, which is closer to the 3'-end of the genome) is more active (reference: "Properties and use of novel replication-competent vectors based on Semliki Forest virus. Kai Rausalu, Anna Iofik, Liane Ülper, Liis Karo-Astover, Valeria Lulla & Andres Merits. Virology Journal volume 6, Article number: 33 (2009)"). In the PIV constructs described in our article, the second copy (SG2) directs RNA synthesis for structural proteins, while the first copy (SG1) controls the gene cassette containing GFP and SEAP genes. Therefore, it is reasonable to expect that the activity of SG1 in PIV will be lower than that of SG2, and both promoters in PIV will be less active than the SG promoter in replicons.
- Reviewer 1 wrote: … Another important feature of alphavirus-based replicons, is their biosafety, and this issue should be also, in my opinion, experimentally addressed and described to convince that the PIV system is practical for general use in laboratories. Using BSL-2 conditions would increase the costs of production. The strategy of PIV with RH-piv helper seems to increase the hazard of replication-competent virus (RCV) generation (as the authors admit, during vector production, a mix of PIV and RH-piv particles is expected). The authors used the VEEV vaccine strain TC-83 with reduced pathogenicity; however, to propose BSL-1 conditions to this system, the authors should follow the existing regulations for amplicon systems, providing information on formation of RCV during PIV production (tested by appropriate assays).
Authors’ response: The biosafety of the system is, of course, a crucial concern. To address this, we conducted additional experiments using PIV samples packaged by cotransfection with the RH-piv helper. This experiment involved four blind passages on permissive HEK293FT cells, followed by detection of the virus in the conditioned medium collected during these passages. The experimental conditions and results are detailed in the Materials and Methods (lines 400-420), Results (lines 594-605), and Discussion (lines 772-782). The findings show that the bipartite system consisting of PIV and RH-piv can undergo a limited number of infectious passages in culture. However, the number of such passages is restricted, and the presence of vector particles decreases with successive infectious passages. No virus capable of self-sustained propagation was detected.
- Reviewer 1 wrote: Using BSL-2 conditions would increase the costs of production. Propose BSL-1 conditions to this system, the authors should follow the existing regulations…
Authors’ response: In response to this comment, we would like to address the following points:
- We conduct our work in BSL-2 conditions and are based in an academic institute.
- Nearly all laboratories in academic settings operate at BSL-2.
- The TC-83 virus, from which the PIV vector and helper are derived, is approved for work in BSL-2 laboratories (see reference below for source of information).
- The PIV vectors for recombinant expression described in our study were developed as an exemplary technology for efficient recombinant protein production in academic laboratories. This technology overcomes challenges common in academic settings, which are not typically encountered in industry environments.
The majority of biological laboratories in academia worldwide are certified as Biosafety Level 2 (BSL-2), as university research often involves moderate-risk biological agents that require precautions beyond BSL-1 but do not need the stringent containment measures of BSL-3 or BSL-4. Our lab is also BSL-2, and we do not propose working with TC-83 (the parent virus of our constructs) in BSL-1 conditions.
Importantly, the TC-83 virus is considered a moderate-risk biological agent and is approved for use in BSL-2 conditions. Reference: "Guidelines for Research Involving Viral Vectors: Alphaviruses and Alphavirus Vectors." (https://www.southalabama.edu/departments/research/compliance/biosafety/resources/alphavirus.vector.guidelines.pdf). Citation: “VEEV strain TC-83 are classified as RG-2 pathogens and may be manipulated using BSL-2 facilities and work practices”.
Although our technique (PIV + PH) is primarily suited for academic settings, it would still comply with current regulations if applied in the industry, as biotech industry facilities are typically certified at BSL-2 or higher. This is due to the generally stricter regulations in the industrial sector compared to academia.
- Reviewer 1 wrote: The experiment described in Figure 5 should, have a negative control, without the RH-piv construct, to assess the ability of self-transmission of the VEEV genome.
Authors’ response: The experiment suggested by the Reviewer was carried out. To evaluate the ability of the PIV genome to produce infectious particles without the helper, cells were transfected with PIV alone, and titers were measured. It was found that PIV generates small quantities of infectious particles, with titers not exceeding 1000 infectious particles per milliliter. This titer is five orders of magnitude lower than the titers achieved in the presence of the helper. Therefore, the limited ability of PIV to self-propagate will not impact the production of the recombinant product. The results are presented in new Figure 5 and in lines 588-593.
- Reviewer 1 wrote: Another major remark regards the conclusion on IFN-dependance of expression. In my opinion, such a conclusion cannot be drawn when comparing two cell lines originating from distinct species… sustain this conclusion, they should compare HEK293FT cells with human IFN-deficient cells. “of note” remark during discussion, that the tested cell lines differ in their responsiveness to IFNs.
Authors’ response: In the article, we do not conclude that expression levels are dependent on interferon (IFN) or on the ability of the cell line to establish an IFN-induced antiviral state. Instead, we discuss a different point: the production correlates with the presence of cytopathic effect (CPE), which, in the case of alphaviruses, is caused by the specific cytopathic activity of the capsid protein. In our case, with the vectors described in the article, CPE occurs when the helper expressing the capsid protein is present in the cell.
The reason we used two cell lines is unrelated to studying the effects of interferon; we did not set out to explore the impact of interferon on expression. The reason for using two cell lines is rather historical: we selected BHK-21 because this cell line is frequently used in alphavirus research, and a large body of studies on alphavirus (virus, replicon) replication in BHK-21 exists. It may be of interest to readers to compare our results with those of other groups using BHK-21. The other cell line, HEK293FT, was chosen based on the authors’ prior experience with this cell line, as it provides very high transfection efficiency. In the experiments described in the article, we found that HEK293FT cells also have the advantage of producing significantly more product compared to BHK-21, although the reasons for this increased production remain unclear.
We added a note in the Discussion section mentioning that BHK-21 and HEK293FT cell lines differ in their ability to respond to IFN, in lines 852-857. This mention is important because when infecting IFN-competent cells with viral vectors, one typically needs to consider the effects of IFN.
- Reviewer 1 wrote: Please use Italics for Latin.
Authors’ response: Done.
- Reviewer 1 wrote: ORFs for GFP and SEAP were joined in a contiguous ORF through the 2A-peptide. What was the efficiency of GFP and SEAP separation by the 2A peptide during translation in both tested cell lines, and can it contribute to the active SEAP yield?
Authors’ response: Cleavage efficiency of 2A peptide was not studied in our work because the primary focus was to compare SEAP expression across different constructs. It is important to note that the same gene cassette GFP-2A-SEAP was used in all constructs, and any potential impact of 2A peptide processing on SEAP yield would remain same across all constructs, and is therefore unlikely to confound expression differences between replicons and PIV, and a simple plasmid vector.
- Reviewer 1 wrote: Is it possible that the WB in Figure 7C is presenting in fact a double band (with and without GFP?), would be visible when less material is analyzed? Could anti-GFP blot confirm the efficiency of separation? This is a minor remark, because, of course, the GFP-2A-SEAP construct was used here only as a model. The Western blotting in Figure 7C/Figure S5 seem not to include negative control, e.g. by comparing the lysates used for purification, this would augment the conclusion on band specificity.
Authors’ response: With this Reviewer’s comment we have the following rebuttals: 1. We used anti-His-tag antibody which reveals the presence of a His-tag at the SEAP’s C-terminus. The high specificity of the commercial anti-His-tag antibody used in our study is well-documented by the manufacturer Invitrogen.
- Also, the calculated molecular masses of expression products SEAP (processed secretory protein) and GFP-2A-SEAP (intracellular unprocessed polyprotein) are so different that these proteins cannot run as close bands in SDS-PAGE (the molecular mass for SEAP-10xHis is 57 kDa based on amino-acid sequence and not considering glycosylation. The calculated molecular mass of GFP-2A-SEAP is 88.4 kDa).
Regarding the possibility of a double band, we consider it highly unlikely that a 2A-SEAP fragment lacking GFP is generated and secreted. The SEAP signal peptide, which directs the protein to the secretory pathway, is located downstream of the 2A peptide. Signal peptides are known to be cleaved off upon translocation of a translation product into the endoplasmic reticulum, thus any partially processed product lacking GFP, if anyhow enter the endoplasmic reticulum, will be processed at the C-terminus of signal peptide and generate mature SEAP. Even more likely, such fragments (2A-SEAP) will not enter the secretory pathway efficiently. Therefore, the observed SEAP band likely represents fully processed SEAP, and its different glycosylation forms.
We acknowledge the reviewer’s suggestion regarding negative controls and appreciate their perspective in future continuation of the studies. While lysate controls were not included in Figure 7C, the banding pattern observed is consistent with SEAP expression and secretion. Given the high specificity of the SEAP antibody and the expected efficiency of 2A peptide cleavage, we believe additional blots, including anti-GFP detection, would not significantly alter our conclusions.
Reviewer 2 Report
Comments and Suggestions for Authors
In this manuscript, the authors describe a system for transient gene expression using single-round, replication-incompetent pseudoinfectious viruses (PIVs) based upon Venezuelan equine encephalitis virus (VEEV) vaccine strain TC-83. The general advantage of PIVs is that they are able to infect a broader range of cell types than most transfection techniques and are less susceptible to variation in efficiency due to plasmid quality or cell number, and similar concepts are employed for other viruses such as lentiviruses. In this particular system, HEK-293FT cells are transfected with a construct containing the gene of interest (GOI, here secreted alkaline phosphatase), genomic RNA packaging signal, and replicase proteins together with another construct expressing just the structural proteins but lacking specific packaging signals, which is then used to infect a large number of cells to generate protein of the GOI. The authors also tested two common producer cells lines and mutations in the PIV genes to determine impacts on host shutoff and cytotoxicity, and found that the wild-type viral sequences lead to higher yields of the GOI despite cell death or potential interferon responses.
This particular system has advantages of high viral titers, scalability, and speed, and the authors described well potential troubleshooting steps for the final purification via His tags. Generally speaking, the paper is well written and information clearly presented,. One point for consideration is the higher levels of virus initially produced in 293FT cells vs. BHK: do the genomic and helper constructs have an SV40 origin of replication? If so, the large T antigen expressed by the cells (the T in 293T) would replicate plasmids and maintain higher copy number in 293FT but not BHK.
Some minor formatting changes would also improve a few aspects of the text:
Combining the text descriptions of the figures (the A)..., B)..., etc.) into the smaller text legends directly beneath the figures.
Italicize in vivo and in vito throughout
Moving the 2 to the appropriate superscript / subscript for CO2, Cl2, cm2, throughout the document
Lines 351-353: replacing the u in micrograms with the letter mu as done above in the text
Author Response
- Reviewer 2 wrote: “One point for consideration is the higher levels of virus initially produced in 293FT cells vs. BHK: do the genomic and helper constructs have an SV40 origin of replication? If so, the large T antigen expressed by the cells (the T in 293T) would replicate plasmids and maintain higher copy number in 293FT but not BHK”.
Authors’ response: Our constructs (replicons and PIVs) function as autonomously replicating RNA molecules, derived from a modified RNA-virus genome. These RNA vectors are produced in cells following transfection of DNA-launched molecular infectious clones (MICs) with the nuclear transcription process. However, once transcribed and exported to the cytoplasm, replication and maintenance (of RNA-vectors) in the cytoplasm occurs as RNA replication, independent of nuclear transcription and the plasmid’s capability of replication. Once the RNA replicates in the cytoplasm, its amplification and subsequent recombinant protein production are dictated by the viral RNA replication machinery rather than by the plasmid backbone, including the presence or absence of the SV40 origin.
It is true that our DNA-launched molecular infectious clones (plasmids used to rescue RNA-vectors) contain the SV40 origin of replication (it was inherited from the assembly vector). But again, it is unlikely that this feature significantly affects the replication efficiency of the cytoplasmic RNA genomes. Therefore, with our RNA vectors, the difference in titers of packaged RNA genomes between 293FT and BHK cells is most likely independent of the presence of the SV40 origin.
- Reviewer 2 wrote: “Some minor formatting changes: Combining the text descriptions of the figures (the A)..., B)..., etc.) into the smaller text legends directly beneath the figures”.
Authors’ response: Done.
- Reviewer 2 wrote: Italicize in vivo and in vitro throughout.
Authors’ response: Done.
- Reviewer 2 wrote: Moving the 2 to the appropriate superscript / subscript for CO2, Cl2, cm2, throughout the document
Authors’ response: Done.
- Reviewer 2 wrote: Lines 351-353: replacing the u in micrograms with the letter mu as done above in the text”.
Authors’ response: Done.
Round 2
Reviewer 1 Report
Comments and Suggestions for Authors
I appreciate the explanations and discussion over my comments and suggestions. There are still two important issues that were misunderstood, and I have to ask the authors for revisions:
As regards the biosafety of pseudoinfectious viruses: I did not propose working with the system at BSL-1 conditions, I am very well aware about the classification of the viral vectors. My comment was about the introduction of the system to the industry, as this is industry that eventually commercializes vectors and vaccines. VEEV vectors are classified as BSL3 or BSL2 depending on their ability to produce infectious particles. The authors rightly performed a study evaluating the infectivity potential and it suggested BSL2 conditions as sufficient. However, all BSL conditions higher that BSL1 increase the final costs of production. What is feasible for academic purposes, may not be as attractive to the industry, if the company has to choose between the BSL1 and BSL2-based production. I would like to ask the authors for a small comment covering this issue, that their vector is a BSL2 agent, that it is suitable for academic research, but may affect the costs of commercialization.
As regards the dependance of IFN, we have another misunderstanding. If, indeed, the use of cell lines is unrelated to studying the effects of interferon, then the title of paragraph 3.1. Titers and recombinant protein production are higher for IFN-competent cells, and higher for the replicon with wild-type replicase, in the first part is not sufficiently supported, as no human IFN-incompetent cell line was used in the study. Please change the title, please justify the choice of cell lines described in lines 494-496 in another way, not suggesting the effect of IFN. In this study, it is an observation, that the IFN-competent human cell line provided better conditions, but it should not lead to far-going conclusions.
Author Response
- Reviewer 1 wrote: My comment was about the introduction of the system to the industry. VEEV vectors are classified as BSL3 or BSL2 depending on their ability to produce infectious particles… However, all BSL conditions higher that BSL1 increase the final costs of production. What is feasible for academic purposes, may not be as attractive to the industry, if the company has to choose between the BSL1 and BSL2-based production. I would like to ask the authors for a small comment covering this issue, that their vector is a BSL2 agent, that it is suitable for academic research, but may affect the costs of commercialization.
Authors’ response: We understand this reviewer's concerns, as industrial producers often maintain their production conditions at the BSL-1 level, which is less costly than BSL-2. In response to the reviewer's suggestion, we have added a new paragraph in the Discussion section (lines 779-793). This paragraph highlights that introducing the VEEV vector into industrial production would lead to increased costs if the industrial entity is required to upgrade to higher BSL.
Here is the added paragraph which addresses the suggestion:
“This study introduces a system for efficient TGE in academic labs. TGE is well-suited for research due to its speed (days to weeks vs. months for stable lines), flexibility (one cell line, standardized protocol for various proteins), and ability to produce small to moderate protein amounts without requiring bioreactors. While transfection-based TGE is used in industry for early-stage development, it is less suitable for large-scale biomanufacturing due to short-lived expression, high transfection costs, and lower productivity compared to stable cell lines, which, despite their longer development time, are optimized for high-yield production and can surpass TGE in efficiency.
Based on the results of this study, how feasible is it to introduce this system into in-dustry? Our vector is classified as BSL-2, making it suitable for academic research but potentially increasing commercialization costs for industrial producers, who typically prefer more cost-effective BSL-1 systems. Transitioning to higher biosafety levels (BSL-2 and above) is expected to raise costs, facility requirements, and regulatory complexity, which could make even efficient split-genome VEEV vectors less appealing for industrial commercialization.”.
- Reviewer 1 wrote: If the use of cell lines is unrelated to studying the effects of interferon, then the title of paragraph 3.1. Titers and recombinant protein production are higher for IFN-competent cells, and higher for the replicon with wild-type replicase” in the first part is not sufficiently supported, as no human IFN-incompetent cell line was used in the study. Please change the title
Authors’ response: The title of paragraph 3.1. was changed to: “3.1. Titers and recombinant protein production using replicon vectors” (line 481).
- Reviewer 1 wrote: Please justify the choice of cell lines described in lines 494-496 in another way, not suggesting the effect of IFN. In this study, it is an observation, that the IFN-competent human cell line provided better conditions, but it should not lead to far-going conclusions.
Authors’ response: This paragraph was completely rewritten to exclude any implication that the selection of cell lines was intended to study the effects of interferon (IFN). The revised text is located in lines 483-496.
Here is the corrected paragraph:
“This study utilized two cell lines, BHK-21 and HEK293FT, selected based on published experience in alphavirus research. The BHK-21 cell line is widely used in alphavirus studies [30, 22, 31], primarily because it cannot suppress viral replication and supports persistent infection [32-33]. HEK293FT was chosen for its high chemical transfection efficiency, confirmed in authors’ prior work [34]. However, it only supports the acute phase of alphavirus infection and inhibits VEEV replication in long-term experiments [35]. This resistance is because HEK293FT cells establish an antiviral state that depends on IFN-I signaling. Although this study did not aim to investigate the impact of the antiviral state on expression, it was of interest to compare vectors with varying sensitivity to the antiviral response in HEK293FT cells. To this end, two different replicon vectors were designed and tested. The literature describes mutations in nsP2 or nsP3 of VEEV that enable replicons to replicate indefinitely in IFN-I-competent cells like HEK293FT [25]. A Q739L mutation [25] was introduced into nsP2 in one replicon, while the other retained the wild-type nsP2”.
